# Synthesis of Novel Sulfamethaoxazole 4-Thiazolidinone Hybrids and Their Biological Evaluation

**DOI:** 10.3390/molecules25163570

**Published:** 2020-08-06

**Authors:** Mashooq A. Bhat, Mohamed A. Al-Omar, Ahmed M. Naglah, Azmat Ali Khan

**Affiliations:** 1Department of Pharmaceutical Chemistry, College of Pharmacy, King Saud University, Riyadh 11451, Saudi Arabia; malomar1@ksu.edu.sa (M.A.A.-O.); azkhan@ksu.edu.sa (A.A.K.); 2Department of Pharmaceutical Chemistry, Drug Exploration and Development Chair (DEDC), College of Pharmacy, King Saud University, Riyadh 11451, Saudi Arabia; anaglah@ksu.edu.sa; 3Peptide Chemistry Department, Chemical Industries Research Division, National Research Centre, Dokki, Cairo 12622, Egypt

**Keywords:** 4-thiazolidinones, sulfamethaoxazole, antimycobacterial activity, cytotoxicity study

## Abstract

A search for potent antitubercular agents prompted us to design and synthesize sulfamethaoxazole incorporated 4-thiazolidinone hybrids (**7a**–**l**) by using a cyclocondensation reaction between 4-amino-*N*-(5-methylisoxazol-3-yl)benzenesulfonamide (**4**), aryl aldehyde (**5a**–**l**), and mercapto acetic acid (**6**) resulting in good to excellent yields. All the newly synthesized 4-thiazolidinone derivatives were screened for their in vitro antitubercular activity against *M. Bovis BCG* and *M. tuberculosis H37Ra* (*MTB)* strains. The compounds **7d**, **7g**, **7i**, **7k**, and **7l** revealed promising antimycobacterial activity against *M. Bovis* and *MTB* strains with IC_90_ values in the range of 0.058–0.22 and 0.43–5.31 µg/mL, respectively. The most active compounds were also evaluated for their cytotoxicity against MCF-7, HCT 116, and A549 cell lines and were found to be non-cytotoxic. Moreover, the synthesized compounds were also analyzed for ADME (absorption, distribution, metabolism, and excretion) properties and showed potential as good oral drug candidates.

## 1. Introduction

Tuberculosis (TB), caused by the pathogen *Mycobacterium tuberculosis* (*MTB*) is one of the most infectious causes of mortality worldwide [1]. The pathogenic synergy between TB and HIV is alarming. Moreover, TB frequently occurs in human immunodeficiency virus (HIV)/AIDS patients [2,3]. In 2019, according to the World Health Organization (WHO), 1.5 million deaths were reported due to TB, out of which 0.36 million people were infected with both HIV and TB. Worldwide, the proportion of new cases with multidrug-resistant tuberculosis (MDR-TB) was 3.5% and has not changed in comparison to recent years [4]. Bacillus Calmette-Guerin (BCG) is an attenuated derivative of a virulent strain of *Mycobacterium bovis* which has been used as a vaccine against *MTB*, as a recombinant vehicle for multivalent vaccines against other infectious diseases, and as cancer immunotherapy [5]. Therefore, the ability to rapidly and specifically identify BCG is clinically important. In addition to this, totally drug-resistant TB (TDR-TB) has recently arisen which is resistant to all clinical drugs [6]. Delamanid (OPC-67683) and Bedaquiline (TMC207) are the two drugs recommended by the US FDA for multi-drug-resistant tuberculosis (MDR-TB) treatment [7,8]. However, at present there are no medicinal drugs under clinical trials. Because of this, there is an urgent need for novel, safe, and effective antimycobacterial drugs which will efficiently treat XDR and MDR tuberculosis.

Numerous heterocyclic compounds containing nitrogen, sulfur, and oxygen heteroatoms have attracted the attention of chemists over the years because of their biological properties. In particular, the thiazolidin-4-ones, a privileged pharmacophore possesses an array of biological activities including anti-inflammatory [9], antimicrobial [10], antibiofilm [11], antiparasitic [12], anticancer [13], anti-viral [14], antidiabetic [15], antiarthritic [16], FSH agonist [17], JNK stimulating phosphatase-1 (JSP-1) [18], CNS-penetrant [19], CDK1/Cyclin B inhibition [20], COX-2 inhibitor [21] and hypoglycemia [22] properties. In the last few years, Kucukguzel et al. developed substituted 4-thiazolidinones and among them compounds **1** and **4** exhibited excellent antitubercular activity with 90% and 98% inhibitions at 6.25 µg/mL, respectively [23]. Jaju and co-workers reported antitubercular activity of isonicotinyl hydrazide derivatives against *M. tuberculosis H37Rv* strain. Among the synthesized isonicotinylhydrazide derivatives compound **3** was found to be more active with MIC = 0.31 µg/mL [24]. Recently, compounds **2**, **5**, and **6** bearing 4-thiazolidinone scaffolds were reported for their excellent antitubercular activities [25,26,27,28,29,30]. Some representative biologically active 4-thiazolidinones are shown in Figure 1.

A literature survey revealed that sulfonamide derivatives are endowed with numerous therapeutic activities as anticonvulsants, HIV protease inhibitors, and antitumor agents [31]. Sulfonamide derivatives are also used for treating certain forms of malaria, urinary tract infections, erectile dysfunction [32] as well as use as herbicides and dyes [33]. Isoxazole derivatives also possess various interesting pharmacological properties. 5-methylisoxazole is present in many marketed drugs such as Valdecoxib, Parecoxib, and Leflunomide and exhibits broad spectrum activity such as anticancer, antiviral, antibacterial, analgesic, anticonvulsant, antinociceptive, and immunomodulating activities [34].

The design of sulfamethaoxazole 4-thiazolidinone hybrids requires the consideration of three different parts, as depicted in Figure 2. The 4-thiazolidinone motif is the main backbone of the design strategy. It helps to enhance the pharmacophoric properties of the synthesized compounds [35]. The second segment is the sulfonamide unit linked to the isoxazole nucleus which is responsible for the biological activity. Sulfonamides and isoxazole scaffolds are present in many reported drugs such as Celecoxib and Parecoxib. Last of the all, the aryl ring with the diverse substitutional unit, is responsible for the lipophilicity control while contributing as a highly potent pharmacological part due to the existence of various functional groups.

According to the pharmacological importance of the above reported compounds and in continuation of our efforts based on the design, synthesis and biological evaluation of heterocyclic derivatives [36,37], we designed and synthesized 4-thiazolidinone hybrids by assembling, in a single molecular framework, sulfonamides and isoxazole units, with the aim to obtain prominent antimycobacterial activity with minimal side effects.

## 2. Results and Discussion

The protocol for the syntheses of a series of new sulfamethaoxazole 4-thiazolidinone hybrids (**7a**–**l**) is reported in Scheme 1 and Scheme 2. Compounds (**7a**–**l**) were prepared from a cyclocondensation reaction between 4-amino-*N*-(5-methylisoxazol-3-yl)benzenesulfonamide (**4**), aryl aldehydes (**5a**–**l**), and mercapto acetic acid (**6**) (Scheme 2).

The required starting materials, 4-amino-*N*-(5-methylisoxazol-3-yl)benzenesulfonamide (**4**) were prepared from commercially available *N*-acetylsulfanilyl chloride (**1**) and 3-amino-5-methylisoxazole (**2**) to give the *N*-acylated intermediate (**3**) followed by acid hydrolysis [38] and details as depicted in (Scheme 1). The synthesis of new sulfamethaoxazole-incorporated 4-thiazolidinone hybrids (**7a**–**l**) was achieved via a cyclocondensation reaction between 4-amino-*N*-(5-methylisoxazol-3-yl) benzenesulfonamide (**4**), aryl aldehydes (**5a**–**l**), and mercapto acetic acid (**3**) in toluene at reflux temperature for 2 h (Scheme 2).

All the newly synthesized compounds were characterized by ^1^H NMR, ^13^C NMR and HRMS techniques. The ^1^H NMR spectrum of compound **7a**, the methylene protons of the 4-thiazolidinone ring appeared as two distinct doublet of doublets at δ 3.76 and 3.72 ppm, indicating that both the protons were magnetically non-equivalent. The C–H proton of the 4-thiazolidinone ring appeared as a singlet at δ 6.71 ppm. The formation of the 4-thiazolidinone ring was also proven by the appearance of peaks at 33.04 (CH_2_), 68.44 (CH), and 179.2 (C=O) ppm, respectively in the ^13^C NMR spectrum of compound **7a**. The HRMS spectrum further strengthened the structure assigned to (**7a**) as *N*-(5-methylisoxazol-3-yl)-4-(4-oxo-2-phenylthiazolidin-3-yl)benzenesulfonamide, showing [M + H]^+^ ion peak at *m*/*z* 416.2147 for its molecular formula C_19_H_18_N_3_O_4_S. Experimental procedures and spectra of 4-thiazolidinone derivatives (**7a**–**l**) are given in the Appendix A. The synthetic sequence is depicted in Scheme 1.

### 2.1. Anti-Mycobacterial Activity

The newly synthesized sulfamethaoxazole 4-thiazolidinone hybrids (**7a**–**l**) were screened for in vitro antitubercular activity against *M. Bovis BCG* (ATCC 35743) and *MTB H37Ra* (ATCC 25177) in liquid medium [39]. We explored the eminent XTT Reduction Menadione assay (XRMA) of the anti-mycobacterial screening protocol employing first-line anti-mycobacterial rifampicin drug as a standard reference and the IC_50_ and IC_90_ values are presented in Table 1.

The sulfamethaoxazole 4-thiazolidinone derivatives **7d**, **7g**, **7i**, **7k**, and **7l** exhibited excellent antitubercular activities with IC_90_ values ranging from 0.058–0.22 and 0.43–5.31 µg/mL. They are found to be most active against *M. bovis BCG* and *MTB H37Ra* and strain, respectively. However, the rest of the derivatives **7a**, **7b**, **7c**, **7e**, **7f**, **7h** and **7j** were found to be less active against *M. bovis BCG* and *MTB H37Ra* strain with IC_90_ = >30 µg/mL.

### 2.2. Structure Activity Relationship (SAR)

According to the in vitro data, the sulfamethaoxazole incorporated 4-thiazolidinone hybrids displayed significant anti-mycobacterial activity and the results are presented in Table 1. The hybrid molecules derived from 4-amino-*N*-(5-methylisoxazol-3-yl) benzenesulfonamide, aryl aldehydes, and mercapto acetic acid have been used as a basic skeleton for the development of structural analogues, which produced the sulfamethaoxazole incorporated 4-thiazolidinone derivatives (**7a**–**l**). The results of the biological evaluation revealed that the activity was considerably affected by introducing various substituents on the aryl ring.

Firstly, we discuss the anti-mycobacterial activity of synthesized compounds against *M. Bovis BCG* strain. From the series (**7a**–**l**), compound **7a** without any substituent on the aryl ring displays lesser potency with IC_90_ value >30 µg/mL against *M. Bovis BCG* strains as compared to the standard drug rifampicin and the results are displayed in Table 1. Compounds **7b** in which (R_3_ = *–methyl*) and **7c** in which (R_3_ = *−methoxy*) showed lower antitubercular activity against *M. Bovis BCG* strain with IC_90_ values >30 µg/mL. The introduction of the *fluoro* group in compound **7d** (R_3_ = *−fluoro*) presented excellent antitubercular activity against *M. Bovis BCG* strain with IC_90_ value 0.22 μg/mL. Introduction of the chloro group in the aryl ring of compound **7e** (R_1_ = *–chloro*) and **7f** (R_2_ = *–chloro*) showed lesser potency against *M. Bovis BCG* strain with IC_90_ value >30 μg/mL. With a chloro group in the R_3_ position in compound **7g** (R_3_ = *−chloro*) it was highly potent against *M. Bovis BCG* strain with an IC_90_ value 0.13 μg/mL.

Replacing the chloro group by a bromo group **7h** (R_3_ = *–bromo*) exhibits less activity towards *M. Bovis BCG* strain with IC_90_ value >30 µg/mL. Introduction of a CF_3_ group at the para position of the aryl group **7i** (R_3_ = *−CF_3_*) compound displays excellent antitubercular effect against *M. Bovis BCG* with 0.17 µg/mL. With the introduction of a nitro group at the para position **7j** (R_3_ = *–NO_2_*) the compound does not show any antitubercular activity against the *M. Bovis BCG* strain. In compounds **7k** (R_1_ = R_4_ = *–fluoro*) the compound showed a sharp increase in antitubercular activity against the *M. Bovis BCG* strain with 0.15 μg/mL. With the disubstituted compound **7l** (R_1_ = R_4_ = ***–****chloro*) it showed an increase in antitubercular activity with IC_90_ value 0.058 µg/mL against *M. Bovis BCG* strain. Hence, among all the newly synthesized compounds (**7a**–**l**), compounds **7d**, **7g**, **7i**, **7k**, and **7l**, exhibit promising anti-mycobacterial activity against *M. Bovis BCG* and the results are summarized in Table 1.

Now, we also screened the antitubercular activity against the *MTB H37Ra* strain. From the 4-thiazolidinone series (**7a**–**l**), compound **7a** without any substitution on the aryl ring showed lower anti-mycobacterial activity with IC_90_ value >30 µg/mL against *MTB H37Ra* strain and the results are shown in Table 1. Compounds **7b** in which (R_3_ = *–methyl*) and **7c** in which (R_3_ = *–methoxy*) showed lower activity against *MTB H37Ra* strain with IC_90_ values >30 µg/mL. With the introduction of the *fluoro* group, compound **7d** (R_3_ = *−fluoro*) showed prominent activity against *MTB H37Ra* strain with IC_90_ value of 0.70 μg/mL.

Introduction of the chloro group in the aryl ring of compound **7e** (R_1_ = *−chloro*) and **7f** (R_2_ = *–chloro*) resulted in them exhibiting less potency against the *MTB H37Ra* strain with IC_90_ value >30 μg/mL. With a chloro group in R_3_ position in compound **7g** (R_3_ = *−chloro*), it results surprisingly with an increase in antitubercular activity against *MTB H37Ra* strain with IC_90_ value 5.31 μg/mL. Replacing the chloro group by a bromo group **7h** (R_3_ = *–bromo*) results in lower activity with IC_90_ value >30 µg/mL against *MTB H37Ra* strain. Introduction of a CF_3_ group at the para position of the aryl group **7i** (R_3_ = *−CF_3_*) exhibits promising antitubercular against *MTB H37Ra* with IC_90_ value 1.08 µg/mL. Introduction of the nitro group at the para position **7j** (R_3_ = *−NO_2_*) does not show any anti-mycobacterial activity against the *MTB H37Ra* strain. With disubstituted compounds **7k** (R_1_ = R_4_ = *–fluoro*) there is displayed excellent activity against the *MTB H37Ra* strain with 0.71 μg/mL. Compounds **7l** (R_1_ = R_4_ = *–chloro*) show promising anti-mycobacterial activity with IC_90_ value 0.43 µg/mL against *MTB H37Ra* strain. Hence, among all the synthesized compounds **7a**–**l**, compounds **7d**, **7g**, **7i**, **7k**, and **7l** showed excellent antitubercular activity against *MTB H37Ra* and the results are displayed in Table 1.

### 2.3. Cytotoxicity

The highly active sulfamethaoxazole incorporated 4-thiazolidinones **7d**, **7g**, **7i**, **7k**, and **7l** were tested for cytotoxicity activity against three human cancer cell lines MCF-7, HCT 116, and A549 using the well established MTT protocol [40]. The cytotoxicity results of these compounds indicate they are highly potent and are specific inhibitors against *M. Bovis BCG* and *MTB H37Ra* strain with GI_50_/GI_90_ (>100 µg/mL). Thus, all the most active compounds were relatively non-toxic against MCF-7, HCT 116 and A549 cell lines with (GI_50_/GI_90_) of >100 and the results are incorporated in Table 2.

### 2.4. Selectivity Index (SI)

The selectivity index indicates that a highly potent compound is only active against *mycobacteria* but it is non-toxic against host human cell lines. According to a study on the drug susceptibility of TB, antitubercular activity was considered to be specific when the selectivity index was >10 [41]. This study suggested that, compounds **7d**, **7g**, **7i**, **7k**, and **7l** display the highest selectivity index >10, suggesting that these compounds act as highly potent antimycobacterial agents and should be modified for the next level. The selectivity index study results are incorporated in Table 3.

### 2.5. Antibacterial Activity

To determine the specificity of the most potent compounds **7d**, **7g**, **7i**, **7k**, and **7l** were evaluated for their antibacterial activity against Gram-negative bacteria (*P. flurescense* ATCC 13525), (*E. coli* ATCC 25292) and Gram-positive bacteria (*B. subtillus* ATCC 23857), (*S. aureus* ATCC 29213). The antibacterial activity protocol suggests that all the active compounds were much less active towards bacterial strains. All the most active compounds exhibited higher specificity towards *M. Bovis BCG* and *MTB H37Ra* strains (Table 4).

### 2.6. ADME Properties

The success of a drug is determined not only by good efficacy but also by an acceptable ADME profile. In this study, we calculated the molecular volume (MV), the molecular weight (MW), the logarithm of partition coefficient (miLog *P*), the number of hydrogen bond acceptors (n-ON), the number of hydrogen bonds donors (n-OHNH), the topological polar surface area (TPSA), the number of rotatable bonds (n-ROTB) and Lipinski’s rule of five [42] using the Molinspiration online property calculation tool kit [43]. Absorption (%ABS) was calculated by: %ABS = 109 − (0.345 × TPSA) [44].

The drug-likeness model score (a collective property of physico-chemical properties, pharmacokinetics, and pharmacodynamics of a compound, represented by a numerical value) was computed by MolSoft software [45]. A computational study of all the synthesized compounds was performed for prediction of the ADME properties and the values obtained are presented in Table 4. It is apparent that the compounds exhibited a good %ABS (%absorption) ranging from 69.11 to 85.02%.

Furthermore, the compounds **7d**, **7g**, **7i**, **7k**, and **7l** did not violate Lipinski’s rule of five (miLog *P ≤* 5) and results are incorporated in Table 5. A molecule likely to be developed as an orally active drug candidate should show no more than one violation of the following four criteria: miLog *P* (octanol–water partition coefficient) ≤5, molecular weight ≤500, number of hydrogen bond acceptors ≤10, and number of hydrogen bond donors ≤5 [46]. The larger the value of the drug likeness model score, the higher is also the probability that the particular molecule will be active. All the tested compounds followed the criteria for orally active drugs and therefore, these compounds may have a good potential for eventual development as oral agents.

According to the literature survey, it was found that these type of compounds can bind to the matrix metalloproteinase-8 (MMP-8; also known as neutrophil collagenase; PDB ID: 5h8x) [47,48]. This endopeptidase is part of a complex proteolytic MMP enzyme family, where its over expression has been linked to several pathological processes [48]. As several articles had already described the coordination of metals by sulfur containing compounds [49,50], we investigated whether the sulfamethaoxazole in our compounds could coordinate the Zn^2+^ ion in MMP-8. This could promote potential inhibitory activity on MMP-8, as presented by Tauro et al. [48]. According to previous reports of molecular docking studies our target compounds should easily coordinate the Zn^2+^ in the receptor cavity. The interactions of the sulfamethaoxazole moiety was identical to the interactions observed in the original catechol ligand (hydrogen bonds with backbones of Ala161 and Leu160).

## 3. Materials and Methods

### 3.1. Materials

All the solvents and regents were purchased from Sigma Aldrich (Gillingham, UK) and they were used in the reactions without purification. The progress of each reaction was monitored by ascending thin layer chromatography (TLC) using TLC aluminum sheets, silica gel 60F_254_ precoated (Merck, Darmstadt, Germany), and locating the spots using UV light as the visualizing agent or iodine vapor. The melting points for the 4-thiazolidinone conjugates were carried out using an open capillary method and they are uncorrected. ^1^H NMR spectra were obtained on a Bruker DRX-400 MHz NMR spectrometer in DMSO-*d*_6_ (δ in ppm relative to tetramethylsilane (TMS) as an internal standard) and ^13^C NMR spectra were recorded (DMSO-*d*_6_) on a Bruker DRX-100 MHz spectrometer (Bruker BioSpin MRI GmbH, Ettlingen, Germany). High-resolution mass spectra (HRMS) were obtained using an Agilent 6520 (QTOF) ESI-HRMS model (Thermo Fisher Scientific, Waltham, MA, USA).

### 3.2. Typical Experimental Procedure for the Synthesis of 4-Amino-N-(5-methylisoxazol-3-yl)benzenesulfonamide (**4**)

A mixture of *N*-acetylsulfanilyl chloride (**1**) (10 mmol), 3-amino-5-methylisoxazole (**2**) (10 mmol), and K_2_CO_3_ (5 mmol) was dissolved in acetonitrile (30 mL). The reaction mass was refluxed for 2 h to give the *N*-acylation intermediate (**3**) followed by acid hydrolysis. The progress of the reaction was monitored by using TLC. After 3 h, the reaction mass was poured into ice cold water. The obtained solid was filtered and washed with water. The crude solid was crystallized in ethanol to afford the corresponding pure product (**4**) and used for the further reaction without purification.

### 3.3. Typical Experimental Procedure for the Synthesis of Sulfamethaoxazole Incorporated Substituted 4-Thiazolidinone Hybrid (**7a**)

The mixture of appropriate 4-amino-*N*-(5-methylisoxazol-3-yl)benzenesulfonamide (**4**) (1 mmol), benzaldehydes (**5a**) (1 mmol) and mercapto acetic acid (**6**) (1.2 mmol) in toluene (5 mL) was placed in a round bottom flask. The mixture was refluxed at 120 °C for an appropriate time until the completion of the reaction. The progress of the reaction was monitored by TLC using ethyl acetate: hexane as a solvent system. The reaction mixture was quenched with crushed ice and extracted with ethyl acetate (2 × 15 mL). The organic extracts were washed with brine solution (2 × 15 mL) and dried over anhydrous sodium sulphate. The solvent was evaporated under reduced pressure to afford the corresponding crude compounds. The obtained crude compounds were recrystallized using ethanol.

#### 3.3.1. *N*-(5-methylisoxazol-3-yl)-4-(4-oxo-2-phenylthiazolidin-3-yl)benzenesulfonamide (**7a**)

Compound **7a** was obtained from the cyclocondensation reaction between **4**, **5a**, and **6** for 12 h. White solid; m.p.: 220–222 °C; Yield: 84%; ^1^H NMR (400 MHz, DMSO*-d*_6_, *δ* ppm): 9.49 (s, 1H, NH), 8.05–8.04 (m, 2H, Ar-H), 7.33–7.28 (m, 4H, Ar–H), 7.25–7.22 (m, 3H, Ar–H), 6.71 (s, 1H, S–CH–N), 5.88 (s, 1H, –C=CH), 3.76 (d, 1H, CH_2_, *J* = 12 Hz), 3.72 (d, 1H, CH_2_, *J* = 12 Hz) and 2.39 (s, 3H, –CH_3_); ^13^C NMR (100 MHz, DMSO*-d*_6_, *δ* ppm): 179.27, 169.78, 152.98, 143.54, 140.31, 136.81, 132.99, 131.16, 131.08, 129.61, 127.78, 99.76, 68.44, 33.04 and 14.53; HRMS (ESI-qTOF): Calcd. for C_19_H_18_N_3_O_4_S_2_ [M + H]^+^, 416.2166: found: 416.2147.

#### 3.3.2. *N*-(5-methylisoxazol-3-yl)-4-(4-oxo-2-(p-tolyl)thiazolidin-3-yl)benzenesulfonamide (**7b**)

Compound **7b** was obtained from the cyclocondensation reaction between **4**, **5b**, and **6** for 12 h. White solid; m.p.: 234–236 °C; Yield: 82%; ^1^H NMR (400 MHz, DMSO*-d*_6_, *δ* ppm): 9.68 (s, 1H, NH), 8.10–8.09 (m, 2H, Ar-H), 7.33–7.31 (m, 2H, Ar-H), 7.27–25 (m, 2H, Ar-H), 7.19–7.18 (m, 2H, Ar-H), 6.68 (s, 1H, S-CH-N), 5.88 (s, 1H, -C=CH), 3.81 (d, 1H, CH_2_, *J* = 12 Hz), 3.77 (d, 1H, CH_2_, *J* = 12 Hz), 2.43 (s, 3H, -CH_3_) and 2.37 (s, 3H, -CH_3_); ^13^C NMR (100 MHz, DMSO*-d*_6_, *δ* ppm): 173.14, 169.97, 152.76, 141.98, 137.28, 133.23, 130.01, 129.47, 128.55, 128.49, 128.23, 100.11, 66.62, 36.89, 24.97 and 12.16; HRMS (ESI-qTOF): Calcd. for C_20_H_20_N_3_O_4_S_2_ [M + H]^+^, 430.0875: found: 430.0893.

#### 3.3.3. 4-(2-(4-methoxyphenyl)-4-oxothiazolidin-3-yl)-*N*-(5-methylisoxazol-3-yl)benzenesulfonamide (**7c**)

Compound **7c** was obtained from the cyclocondensation reaction between **4**, **5c**, and **6** for 12 h. White solid; m.p.: 242–234 °C; Yield: 84%; ^1^H NMR (400 MHz, DMSO*-d*_6_, *δ* ppm): 9.42 (s, 1H, NH), 8.02–8.01 (m, 2H, Ar-H), 7.25–7.24 (m, 2H, Ar-H), 7.21–7.20 (m, 2H, Ar-H), 6.85–6.84 (m, 2H, Ar-H), 6.56 (s, 1H, S-CH-N), 5.80 (s, 1H, -C=CH), 3.79 (s, 1H, -OCH_3_), 3.72 (d, 1H, CH_2_, *J* = 12 Hz), 3.69 (d, 1H, CH_2_, *J* = 12 Hz) and 2.35 (s, 3H, -CH_3_); ^13^C NMR (100 MHz, DMSO*-d*_6_, *δ* ppm): 178.60, 168.20, 157.10, 143.22, 139.70, 132.71, 130.85, 129.58, 129.37, 127.39, 125.19, 102.19, 66.07, 52.53, 32.06 and 13.08; HRMS (ESI-qTOF): Calcd. for C_20_H_20_N_3_O_5_S_2_ [M + H]^+^, 446.0521: found: 446.0546.

#### 3.3.4. 4-(2-(4-fluorophenyl)-4-oxothiazolidin-3-yl)-*N*-(5-methylisoxazol-3-yl)benzenesulfonamide (**7d**)

Compound **7d** was obtained from the cyclocondensation reaction between **4**, **5d**, and **6** for 12 h. Yellow solid; m.p.: 216–218 °C; Yield: 80%; ^1^H NMR (400 MHz, DMSO*-d*_6_, *δ* ppm): 9.54 (s, 1H, NH), 8.02–8.01 (m, 2H, Ar-H), 7.33–7.32 (m, 2H, Ar-H), 7.27–7.25 (m, 2H, Ar-H), 7.07–7.04 (m, 2H, Ar-H), 6.60 (s, 1H, S-CH-N), 6.18 (s, 1H, -C=CH), 3.77 (d, 1H, CH_2_, *J* = 12 Hz), 3.73 (d, 1H, CH_2_, *J* = 12 Hz) and 2.39 (s, 3H, -CH_3_); ^13^C NMR (100 MHz, DMSO*-d*_6_, *δ* ppm): 176.31, 168.22, 161.79, 160.24, 154.29, 143.81, 143.36, 129.57, 129.52, 129.36, 127.42, 125.69, 115.63, 105.09, 68.12, 33.58 and 12.70; HRMS (ESI-qTOF): Calcd. for C_19_H_17_FN_3_O_4_S_2_ [M + H]^+^, 434.0858: found: 434.0893.

#### 3.3.5. 4-(2-(2-chlorophenyl)-4-oxothiazolidin-3-yl)-*N*-(5-methylisoxazol-3-yl)benzenesulfonamide (**7e**)

Compound **7e** was obtained from the cyclocondensation reaction between **4**, **5e**, and **6** for 12 h. White solid; m.p.: 240–242 °C; Yield: 84%; ^1^H NMR (400 MHz, DMSO*-d*_6_, *δ* ppm): 9.62 (s, 1H, NH), 8.27–8.26 (m, 2H, Ar-H), 7.48–7.46 (m, 3H, Ar-H), 7.37–7.31 (m, 3H, Ar-H), 6.87 (s, 1H, S-CH-N), 6.35 (s, 1H, -C=CH), 3.92 (d, 1H, CH_2_, *J* = 12 Hz), 3.90 (d, 1H, CH_2_, *J* = 12 Hz) and 2.23 (s, 3H, -CH_3_); ^13^C NMR (100 MHz, DMSO*-d*_6_, *δ* ppm): 171.64, 167.47, 156.19, 143.33, 141.16, 138.58, 135.94, 132.27, 130.80, 129.46, 127.47, 125.43, 122.06, 101.90, 65.58, 35.18 and 14.01; HRMS (ESI-qTOF): Calcd. for C_19_H_17_ClN_3_O_4_S_2_ [M + H]^+^, 450.0786: found: 450.0796.

#### 3.3.6. 4-(2-(3-chlorophenyl)-4-oxothiazolidin-3-yl)-*N*-(5-methylisoxazol-3-yl)benzenesulfonamide (**7f**)

Compound **7f** was obtained from the cyclocondensation reaction between **4**, **5f**, and **6** for 12 h. White solid; m.p.: 228–230 °C; Yield: 78%; ^1^H NMR (400 MHz, DMSO*-d*_6_, *δ* ppm): 9.56 (s, 1H, NH), 8.04–8.02 (m, 2H, Ar-H), 7.35 (s, 1H, Ar-H), 7.27–7.25 (m, 4H, Ar-H), 7.14–7.11 (m, 1H, Ar-H), 6.65 (s, 1H, S-CH-N), 6.46 (s, 1H, -C=CH), 3.78 (d, 1H, CH_2_, *J* = 12 Hz), 3.74 (d, 1H, CH_2_, *J* = 12 Hz) and 2.40 (s, 3H, -CH_3_); ^13^C NMR (100 MHz, DMSO*-d*_6_, *δ* ppm): 179.77, 167.82, 156.82, 143.99, 143.52, 133.04, 131.60, 129.69, 129.53, 127.56, 125.86, 124.57, 122.12, 103.76, 68.63, 34.98 and 13.79; HRMS (ESI-qTOF): Calcd. for C_19_H_17_ClN_3_O_4_S_2_ [M + H]^+^, 450.1372: found: 450.1396.

#### 3.3.7. 4-(2-(4-chlorophenyl)-4-oxothiazolidin-3-yl)-*N*-(5-methylisoxazol-3-yl)benzenesulfonamide (**7g**)

Compound **7g** was obtained from the cyclocondensation reaction between **4**, **5g**, and **6** for 12 h. White solid; m.p.: 245–247 °C; Yield: 80%; ^1^H NMR (400 MHz, DMSO*-d*_6_, *δ* ppm): 9.76 (s, 1H, NH), 8.02–8.01 (m, 2H, Ar-H), 7.33–7.32 (m, 2H, Ar-H), 7.24–7.23 (m, 2H, Ar-H), 7.19–7.17 (m, 2H, Ar-H), 6.60 (s, 1H, S-CH-N), 6.43 (s, 1H, -C=CH), 3.76 (d, 1H, CH_2_, *J* = 12 Hz), 3.73 (d, 1H, CH_2_, *J* = 12 Hz) and 2.38 (s, 3H, -CH_3_); ^13^C NMR (100 MHz, DMSO*-d*_6_, *δ* ppm): 182.76, 165.85, 157.74, 141.75, 139.99, 128.90, 128.41, 127.88, 124.50, 124.32, 124.19, 99.96, 66.56, 31.80 and 14.15; HRMS (ESI-qTOF): Calcd. for C_19_H_17_ClN_3_O_4_S_2_ [M + H]^+^, 450.0042: found: 450.0025.

#### 3.3.8. 4-(2-(4-bromophenyl)-4-oxothiazolidin-3-yl)-*N*-(5-methylisoxazol-3-yl)benzenesulfonamide (**7h**)

Compound **7h** was obtained from the cyclocondensation reaction between **4**, **5h**, and **6** for 12 h. Red solid; m.p.: 234–236 °C; Yield: 80%; ^1^H NMR (400 MHz, DMSO*-d*_6_, *δ* ppm): 9.64 (s, 1H, NH), 8.10–8.09 (m, 2H, Ar-H), 7.48–7.46 (m, 3H, Ar-H), 7.24–7.22 (m, 2H, Ar-H), 7.17–7.15 (m, 2H, Ar-H), 6.63 (s, 1H, S-CH-N), 6.49 (s, 1H, -C=CH), 3.76 (d, 1H, CH_2_, *J* = 12 Hz), 3.72 (d, 1H, CH_2_, *J* = 12 Hz) and 2.39 (s, 3H, -CH_3_); ^13^C NMR (100 MHz, DMSO*-d*_6_, *δ* ppm): 175.95, 167.58, 156.57, 143.74, 143.29, 135.76, 129.44, 129.19, 125.61, 124.31, 123.96, 97.85, 65.68, 34.06 and 12.32; HRMS (ESI-qTOF): Calcd. for C_19_H_17_BrN_3_O_4_S_2_ [M + H]^+^, 495.1038: found: 495.1071.

#### 3.3.9. *N*-(5-methylisoxazol-3-yl)-4-(4-oxo-2-(4-(trifluoromethyl)phenyl)thiazolidin-3-yl)benzenesulfonamide (**7i**)

Compound **7i** was obtained from the cyclocondensation reaction between **4**, **5i**, and **6** for 12 h. White solid; m.p.: 225–227 °C; Yield: 80%; ^1^H NMR (400 MHz, DMSO*-d*_6_, *δ* ppm): 9.33 (s, 1H, NH), 8.04–8.02 (m, 2H, Ar-H), 7.52–751 (m, 2H, Ar-H), 7.25–7.23 (m, 2H, Ar-H), 6.70 (s, 1H, S-CH-N), 6.06 (s, 1H, -C=CH), 3.75 (d, 1H, CH_2_, *J* = 12 Hz), 3.71 (d, 1H, CH_2_, *J* = 12 Hz) and 2.36 (s, 3H, -CH_3_); ^13^C NMR (100 MHz, DMSO*-d*_6_, *δ* ppm): 178.60, 165.55, 158.09, 142.35, 139.95, 135.99, 135.88, 131.22, 130.49, 130.23, 128.74, 128.05, 98.29, 66.37, 34.75 and 12.43; HRMS (ESI-qTOF): Calcd. for C_20_H_17_F_3_N_3_O_4_S_2_ [M + H]^+^, 484.0728 found: 484.0756.

#### 3.3.10. *N*-(5-methylisoxazol-3-yl)-4-(2-(4-nitrophenyl)-4-oxothiazolidin-3-yl)benzenesulfonamide (**7j**)

Compound **7j** was obtained from the cyclocondensation reaction between **4**, **5j**, and **6** for 12 h. Yellow solid; m.p.: 246–248 °C; Yield: 74%; ^1^H NMR (400 MHz, DMSO*-d*_6_, *δ* ppm): 9.64 (s, 1H, NH), 8.23–8.21 (m, 2H, Ar-H), 8.03–8.01 (m, 2H, Ar-H), 7.55–7.53 (m, 2H, Ar-H), 7.35–7.34 (m, 2H, Ar-H), 6.71 (s, 1H, S-CH-N), 6.15 (s, 1H, -C=CH), 3.78 (d, 1H, CH_2_, *J* = 12 Hz), 3.75 (d, 1H, CH_2_, *J* = 12 Hz) and 2.40 (s, 3H, -CH_3_); ^13^C NMR (100 MHz, DMSO*-d*_6_, *δ* ppm): 179.08, 165.76, 158.84, 141.54, 139.90, 136.65, 134.01, 133.96, 130.38, 128.27, 127.80, 99.96, 67.36, 34.90 and 14.60; HRMS (ESI-qTOF): Calcd. for C_19_H_17_N_4_O_6_S_2_ [M + H]^+^, 461.0075: found: 461.0099.

#### 3.3.11. 4-(2-(2,6-difluorophenyl)-4-oxothiazolidin-3-yl)-*N*-(5-methylisoxazol-3-yl)benzenesulfonamide (**7k**)

Compound **7k** was obtained from the cyclocondensation reaction between **4**, **5k**, and **6** for 12 h. White solid; m.p.: 224–226 °C; Yield: 81%; ^1^H NMR (400 MHz, DMSO*-d*_6_, *δ* ppm): 9.50 (s, 1H, NH), 8.03–8.01 (m, 2H, Ar-H), 7.29–7.27 (m, 2H, Ar-H), 7.22.7.20 (m, 2H, Ar-H), 7.10–7.09 (m, 2H, Ar-H), 6.95 (s, 1H, S-CH-N), 6.45 (s, 1H, -C=CH), 3.85 (d, 1H, CH_2_, *J* = 12 Hz), 3.82 (d, 1H, CH_2_, *J* = 12 Hz) and 2.38 (s, 3H, -CH_3_); ^13^C NMR (100 MHz, DMSO*-d*_6_, *δ* ppm): 174.75, 166.33, 156.87, 141.55, 140.04, 128.82, 128.28, 126.03, 124.64, 123.90, 122.60, 99.67, 67.49, 36.45 and 12.62; HRMS (ESI-qTOF): Calcd. for C_19_H_16_F_2_N_3_O_4_S_2_ [M + H]^+^, 452.1434: found: 452.1481

#### 3.3.12. 4-(2-(2,6-dichlorophenyl)-4-oxothiazolidin-3-yl)-*N*-(5-methylisoxazol-3-yl)benzenesulfonamide (**7l**)

Compound **7l** was obtained from the cyclocondensation reaction between **4**, **5l**, and **6** for 12 h. Yellow solid; m.p.: 218–220 °C; Yield: 80%; ^1^H NMR (400 MHz, DMSO*-d*_6_, *δ* ppm): 9.30 (s, 1H, NH), 8.20 (s. 1H, Ar-H), 8.12–8.11 (m, 2H, Ar-H), 8.01–8.00 (m, 2H, Ar-H), 7.63–7.62 (m, 2H, Ar-H), 7.54–7.51 (m, 2H, Ar-H), 7.31–7.26 (m, 2H, Ar-H), 6.71 (s, 1H, S-CH-N), 6.06 (s, 1H, -C=CH), 3.75 (d, 1H, CH_2_, *J* = 12 Hz), 3.71 (d, 1H, CH_2_, *J* = 12 Hz) and 2.39 (s, 3H, -CH_3_); ^13^C NMR (100 MHz, DMSO*-d*_6_, *δ* ppm): 174.10, 165.46, 155.58, 144.34, 139.34, 130.13, 129.91, 129.82, 127.96, 127.53, 123.76, 123.48, 123.11, 100.51, 65.04, 35.50 and 14.70; HRMS (ESI-qTOF): Calcd. for C_19_H_16_Cl_2_N_3_O_4_S_2_ [M + H]^+^, 485.2672: found: 485.2649.

## 4. Conclusions

In conclusion, the synthesis of sulfamethaoxazole incorporated 4-thiazolidinone hybrids (**7a**–**l**) was carried out and their antitubercular activity against *M. Bovis BCG* and *MTB H37Ra* strains were determined. Among all the tested compounds, **7d**, **7g**, **7i**, **7k**, and **7l** were identified as the most active compounds with activity IC_90_ range 0.22–0.057 against *M. bovis BCG* strain and 0.43–5.31 µg/mL against *MTB H37Ra* strain. The most potent compounds displayed lower cytotoxicity against MCF-7, HCT 116 and A549 cell line using MTT assay, suggesting that these molecules possess highly pharmacodynamic properties. The most active compounds **7d**, **7g**, **7i**, **7k**, and **7l** exhibited a higher selectivity index >10 against MCF-7, HCT 116 and A549 which indicated that they act as prominent antitubercular agents. Thus, this suggests that the compounds from the present series **7d**, **7g**, **7i**, **7k**, and **7l** can be further optimized and developed as lead molecules.

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
