# Peer review of "Synthesis of Novel Sulfamethaoxazole 4-Thiazolidinone Hybrids and Their Biological Evaluation"

_molecules, 2020, doi:10.3390/molecules25163570_

Round 1
Reviewer 1 Report
In the manuscript entitled Synthesis of novel sulfamethaoxazole incorporated substituted 4-thiazolidinone derivatives and their biological, Mashooq Bhat et al. disclosed five substituted 4-thiazolidinone hybrids having promising antitubercular activity, along with no significant cytotoxicity against the cell lines MCF-7, HCT 116 and A549.
The paper is clearly written and the discussion is quite sound and well organized. The results described are interesting, but some issues need to be addressed before the manuscript can be considered for publication:
- INTRODUCTION: Authors should comment more in detail the rationale beyond their drug design: they can also include and discuss the choice of the three different scaffold for new molecule design, adding a literature survay.
- DISCUSSION: The results obtained are interesting, but could be more relevant for this field only after further biological and computational investigations. Initially, authors may propose the potential molecular target(s), using i.e. molecular modelling techniques in order to establish a structural basis for inhibition of M. tuberculosis (a molecular docking study could be carried out against a potential target).
Author Response
Response to Decision Letter
We are extremely grateful to the reviewer(s) for the positive, insightful and constructive comments, which certainly helped to improve the quality of the manuscript. Herewith, we are submitting point to point response to comments/queries posed by the reviewer(s). We have incorporated all the suggestions made by the reviewer(s) in the revised manuscript. The suggestions were very much helpful in shaping the manuscript. We are thankful to the reviewer for their valuable inputs.
The point to point corrections and responses to comments are given below as follows. (Reviewer(s) comments are given in italics and Responses are given in normal font)
Comments and Responses:
Reviewer # 1:
- Comment
The paper is clearly written and the discussion is quite sound and well organized. The results described are interesting, but some issues need to be addressed before the manuscript can be considered for publication
Response
We are extremely grateful to the reviewer for the positive, insightful and constructive comments about our manuscript. Thank you very much for this.
- Comment
INTRODUCTION: Authors should comment more in detail the rationale beyond their drug design: they can also include and discuss the choice of the three different scaffold for new molecule design, adding a literature survay.
Response
We agree with the reviewer comment and suggestion. As per suggestions, we have modified introduction, figure, drug design of synthesis of target molecule and incorporation of relevant references in the revised manuscript.
- Comment
DISCUSSION: The results obtained are interesting, but could be more relevant for this field only after further biological and computational investigations. Initially, authors may propose the potential molecular target(s), using i.e. molecular modelling techniques in order to establish a structural basis for inhibition of M. tuberculosis (a molecular docking study could be carried out against a potential target).
Response
We agree with the reviewer comment and suggestion. As per suggestions, we have studied the ADME properties, drug likeness score and incorporated in the revised manuscript.

Reviewer 2 Report
This paper reports the synthesis, as potential antubercular agents, of a series of sulfamethaoxazole 4-thiazolidinone hybrids, obtained by the well known cyclocondensation reaction of a 4-amino-N-substituted benzenesulfonamide with aryl aldehydes and mercapto acetic acid.
The rationale of the synthetic design relies on the insertion in the same compound of a 4-thiazolidinone and an isoxazole as bioactive units, linked together by a sulfonamide linkage, in order to enhance the potential biological activity of the obtained compounds.
The synthesized derivatives have been tested for in vitro anti-mycobacterial properties against M. Bovis BCG and M. tuberculosis H37Ra strains and some of them showed a moderate biological activity with IC50 values in the range 0.016-0.28 and 0.07-0.16 µg/mL, respectively. The active
compounds were found non-cytotoxic against MCF-7, HCT 116 and A549 cell lines.
The synthetic approach is reliable; however, the structural characterization, based only on the chemical shifts of CH2 protons and the CH proton of 4 thiazolidinone ring as well as the HRMS spectra of the final hybrids 7. I think that the fragmentation pattern of MS spectra could afford useful information.
The main failure of this manuscript is the draft. The english must be completely and carefully revised: whole parts of the paper are scarcely understandable. At my opinion,also the title should be modified for instance in "Synthesis of novel sulfamethaoxazole 4-thiazolidinone hybrids and their biological evaluation".
The paper could be accepted for publication after its complete revision
Author Response
Reviewer # 2:
- Comment
This paper reports the synthesis, as potential an tubercular agents, of a series of sulfamethaoxazole 4-thiazolidinone hybrids, obtained by the well known cyclocondensation reaction of a 4-amino-N-substituted benzenesulfonamide with aryl aldehydes and mercapto acetic acid. The rationale of the synthetic design relies on the insertion in the same compound of a 4-thiazolidinone and an isoxazole as bioactive units, linked together by a sulfonamide linkage, in order to enhance the potential biological activity of the obtained compounds.
Response
We are extremely grateful to the reviewer for the positive, insightful and constructive comments about our manuscript. Thank you very much for this.
- Comment
The synthetic approach is reliable; however, the structural characterization, based only on the chemical shifts of CH2 protons and the CH proton of 4 thiazolidinone ring as well as the HRMS spectra of the final hybrids 7. I think that the fragmentation pattern of MS spectra could afford useful information.
Response
We agree with the reviewer comment and suggestion. In the previous report, for the synthesis of 4-thiazolidinone derivatives are clearly confirmed that on the basis of 1H NMR, 13C NMR and HRMS .please see the relevant references of 4-thiazolidinone derivatives.
- Zeng, F.; Qi, T.; Li, C.; Li, T.; Li, H.; Li, S.; Zhu, L.; Xu, X. Synthesis, structure-activity relationship and binding mode analysis of 4-thiazolidinone derivatives as novel inhibitors of human dihydroorotate dehydrogenase. Chem. Commun., 2017, 8, 1297-1302.
- Deep, A.; Narasimhan, B.; Lim, S. M.; Ramasamy, K.; Mishra, R.K.; Mani, V. 4-Thiazolidinone derivatives: synthesis, antimicrobial, anticancer evaluation and QSAR studies. RSC Adv., 2016, 6, 109485-109494.
- Cheng, P.; Guo, W.; Chen, P.; Liu, Y.; Du, X.; Li, C. an Enantioselective Construction of Chiral Thiazolidine: the asymmetric catalytic [3+2] Annulation of 1,4-dithiane-2,5-diol and ketimines. Commun., 2016, 52, 3418-3421.
- Comment
The main failure of this manuscript is the draft. The english must be completely and carefully revised: whole parts of the paper are scarcely understandable. At my opinion,also the title should be modified for instance in "Synthesis of novel sulfamethaoxazole 4-thiazolidinone hybrids and their biological evaluation".
Response
We apologize for this mistake. We have modified all manuscript, English correction, typological error and title are incorporated in the revised manuscript.

Round 2
Reviewer 1 Report
The authors addressed the concerns of the critiques, adding new biological data and an useful AMDE study, but any hypothesis about potential molecular target(s) was proposed.
Other minor issues are still present in the manuscript (i.e., WHO TB report should updated to 2019, line 51 insert a full stop, line 53 delete the comma, line 58 insert “and” before “among” , line 59 correct exhibits with exhibited, line 61 substitute “them” with “the”, line 141 delete “are”, line 143 insert the full stop and start with “They”, line 144 insert “the”, line 250 insert a space “Table 4” and “the” before compounds etc.).
Line 68: the caption of Figure 1 should be clearly rewritten.
Starting form line 252, the pharagraph should be clearly rewritten because it is understandable.
In particular, the revised part of the manuscript should be carefully checked.
After a new revision, I will recommend publication.
Author Response
Reviewer # 1:
- Comment
The authors addressed the concerns of the critiques, adding new biological data and an useful AMDE study, but any hypothesis about potential molecular target(s) was proposed.
Response
We agree with the reviewer comment and suggestion. We have As per suggestions, we have add the molecular docking study hypothesis of target compounds in the revised manuscript.
According literature survey, it was found that these type of compounds can bind to the matrix metalloproteinase-8 (MMP-8; also known as neutrophil collagenase; PDB ID: 5h8x) [1-2]. This endopeptidase is a part of a very complex proteolytic MMP enzyme family, where its over expression was linked to several pathological processes [2]. As several articles have already described the coordination of metals by sulphur containing compounds [3-4], we investigated whether the sulfamethaoxazole in our compounds could coordinate the Zn2+ ion in the MMP-8. This could promote potential inhibitory activity on MMP-8, as presented by Tauro et al. [2]. According to previous reports of molecular docking studies our target compounds can easily coordinate the Zn2+ in the receptor cavity. The interactions of the sulfamethaoxazole moiety was identical to the interactions observed in the original catechol ligand (hydrogen bonds with backbones of Ala161 and Leu160).
- Bouz, G.; Juhas, M.; Otero, L.P.; Red, C.P.; Jand’ourek, O.; Konecna, K.; Paterova, P.; Kubicek, V.; Janousek, J.; Dolezal, M.; Zitko, J. Molecules 2020, 25, 138.
- Tauro, M.; Laghezza, A.; Loiodice, F.; Piemontese, L.; CaraDonna, A.; Capelli, D.; Montanari, R.; Pochetti, G.; Di Pizio, A.; Agamennone, M. Catechol-based matrix metalloproteinase inhibitors with additional antioxidative activity. Enzym. Inhib. Med. Chem. 2016, 31, 25-37.
- Ogryzek, M.; Chylewska, A.; Turecka, K.; Lesiak, D.; Królicka, A.; Banasiuk, R.; Nidzworski, D.; Makowski, M. Coordination chemistry of pyrazine derivatives analogues of PZA: Design, synthesis, characterization and biological activity. RSC Adv. 2016, 6, 52009-52025.
- Pedersen, K.S.; Perlepe, P.; Aubrey, M.L.; Woodruff, D.N.; Reyes-Lillo, S.E.; Reinholdt, A.; Voigt, L.; Li, Z.; Borup, K.; Rouzieres, M. Formation of the layered
- Comment
Other minor issues are still present in the manuscript (i.e., WHO TB report should updated to 2019, line 51 insert a full stop, line 53 delete the comma, line 58 insert “and” before “among” , line 59 correct exhibits with exhibited, line 61 substitute “them” with “the”, line 141 delete “are”, line 143 insert the full stop and start with “They”, line 144 insert “the”, line 250 insert a space “Table 4” and “the” before compounds etc.).
Response
We agree with the reviewer comment and suggestion. As per suggestions, we have modified above all the suggestion in the revised manuscript.
- Comment
Line 68: the caption of Figure 1 should be clearly rewritten.
Starting form line 252, the pharagraph should be clearly rewritten because it is understandable
Response
We agree with the reviewer comment and suggestion. We have modified and incorporated in the revised manuscript.

Reviewer 2 Report
The manuscript requires further revision for the english lay out. Some examples:
Lines 24-25: "the synthesized compounds were also analyzed for ADME properties and showed potential to build up as good oral drug candidates." could be changed in "the synthesized compounds were also analyzed for ADME properties and showed potential as good oral drug candidates."
Lines 41-42: The phrase "The disease caused by BCG in humans, especially those with cellular immune deficiencies." is unintelligible.
Line 46: "however there is no present medicinal drugs under .." must be changed in "however, there are no present medicinal drugs under .."
Lines 49-52: "There are numerous biologically active molecules with nitrogen, sulphur and oxygen heteroatoms which always drawn the attention of chemist over the years mainly because of their biological importance, therefore, extensive research is still needed to improve their properties and to reduce their adverse effects. Thiazolidin-4-ones a privileged.." could be simply changed in "Numerous heterocyclic compounds containing nitrogen, sulphur and oxygen heteroatoms have attracted the attention of chemists over the years because of their biological properties. In particular, the thiazolidin-4-ones a privileged .."
Lines 55-59: "...Antidiabetic [16], antiarthritic [17], FSH agonist [18], JNK stimulating phosphatase-1 (JSP56 1) [19], colon cancer (HT29) [20], breast cancer (MCF-7) [21], CNS-penetrant [22], CDK1/Cyclin B inhibition [23], COX-2 inhibitor [24] and hypoglycemia [25]. In the last few years, Kucukguzel et al. developed substituted 4-thiazolidinones among them compounds 1 and 4 exhibits excellent antitubercular activity with 90% and 98% inhibitions....." could be changed in "...antidiabetic [16], antiarthritic [17], FSH agonist [18], JNK stimulating phosphatase-1 (JSP56 1) [19], CNS-penetrant [22], CDK1/Cyclin B inhibition [23], COX-2 inhibitor [24] and hypoglycemia [25] properties. In the last few years, Kucukguzel et al. developed 4-thiazoli-dinones 1 and 4 which exhibit excellent antitubercular activity with 90% and 98% inhibitions .."
Lines 69-77: "Literature survey revealed that, sulfonamides derivatives are endowed with numerous therapeutic activities such anticonvulsants, HIV protease inhibitors, and antitumor agents [34]. It is also useful for treating certain forms of malaria, urinary tract infections, herbicides, dyes [35] and erectile dysfunction [36]. Isoxazole and its derivatives received great importance because they possess various interesting pharmacological properties. An isoxazole derivative widely exists in various natural products [37]. 5-methylisoxazole exists in many marketed drugs such as valdecoxib, parecoxib and leflunomide. Methylisoxazole
exhibits broad spectrum activity like anticancer, antiviral, antibacterial, analgesic, anticonvulsant, antinociceptive and immunomodulating activities [38]." could be changed in "Literature survey revealed that sulfonamides derivatives are endowed with numerous therapeutic activities as anticonvulsants, HIV protease inhibitors, and antitumor agents [34]. Sulfonamide derivatives are also used for treating certain forms of malaria, urinary tract infections, erectile dysfunction [36] as well as herbicides and dyes [35]. Also isoxazole derivatives possess various interesting pharmacological properties. 5-methylisoxazole is present
in many marketed drugs such as valdecoxib, parecoxib and leflunomide and exhibits broad spectrum activity like anticancer, antiviral, antibacterial, analgesic,anticonvulsant, antinociceptive and immunomodulating activities [38].
Lines 78-97: "The design of sulfamethaoxazole incorporated substituted 4-thiazolidinones hybrids are mainly divided into three different part as depicted in Figure 2. The 4-thiazolidinone motif is the main backbone of the design strategy. It helps to enhance the pharmacophoric properties as they exhibits drug like properties [39]. The second segment is showing sulfonamides linkages with isoxazole which is responsible for biological activity.Sulfonamides and isoxazole scaffolds are present in many reported drug such as Celecoxib and Parecoxib. Last of all, the aryl ring with the diverse substitutional unit, which is responsible for the lipophilicity control while contributing highly potent pharmacological part due to the existence of various functional groups.
Considering the pharmacological importance of the above and in continuation of our efforts based on the design, synthesis and biological evaluation of heterocyclic compounds [40],
herein, we have design and synthesized 4-thiazolidinone hybrids by accumulating sulfonamides linked with isoxazole unit, substituted aryl rings and mercapto group in a single molecular framework with the hope to obtain prominent antimycobactrial activity with minimal side effects."
I think it is better to change in : "According to the pharmacological importance of the above reported compounds and in continuation of our efforts based on the design, synthesis and biological evaluation of heterocyclic derivatives [40], we have designed and synthesized 4-thiazolidinone hybrids by assembling, in a single molecular framework, sulfonamides and isoxazole units, with the aim to obtain prominent antimycobactrial activity with minimal side effects.
The design of sulfamethaoxazole 4-thiazolidinones hybrids considers three different parts, as depicted in Figure 2. The 4-thiazolidinone motif is the main backbone of the design strategy. It helps to enhance the pharmacophoric properties of the synthesized compounds[39]. The second segment is the sulfonamide unit linked to the isoxazole nucleus which is responsible for the biological activity.Sulfonamides and isoxazole scaffolds are present in many reported drug such as Celecoxib and Parecoxib. Last of all, the aryl ring with the diverse substitutional unit, is responsible for the lipophilicity control while contributing as a highly potent pharmacological part due to the existence of various functional groups."
Lines 100-102: "We have described a protocol for the syntheses of a series of new sulfamethaoxazole incorporated substituted 4-thiazolidinone hybrids (7a-l) as a potential antitubercular agents from commercially available starting materials (Scheme 1 & 2). These compounds (7a-l)..." could be changed in "The protocol for the syntheses of a series of new sulfamethaoxazole 4-thiazolidinone hybrids (7a-l) is reported in Schemes 1 and 2. Compounds 7a-l....."
The above reported examples clearly show that an accurate revision of the text is necessary for the publication of this paper.
Author Response
Response to Decision Letter
We are extremely grateful to the reviewer(s) for the positive, insightful and constructive comments, which certainly helped to improve the quality of the manuscript. Herewith, we are submitting point to point response to comments/queries posed by the reviewer(s). We have incorporated all the suggestions made by the reviewer(s) in the revised manuscript. The suggestions were very much helpful in shaping the manuscript. We are thankful to the reviewer for their valuable inputs.
The point to point corrections and responses to comments are given below as follows. (Reviewer(s) comments are given in italics and Responses are given in normal font)
Comments and Responses:
Reviewer # 2:
- Comment
Lines 24-25: "the synthesized compounds were also analyzed for ADME properties and showed potential to build up as good oral drug candidates." could be changed in "the synthesized compounds were also analyzed for ADME properties and showed potential as good oral drug candidates." Lines 41-42: The phrase "The disease caused by BCG in humans, especially those with cellular immune deficiencies." is unintelligible.
Response
We agree with the reviewer suggestion. We have changed the sentence and given in the revised manuscript.
- Comment
Line 46: "however there is no present medicinal drugs under .." must be changed in "however, there are no present medicinal drugs under .."
Response
We apologize for this mistake. We have modified this correction and incorporated in the revised manuscript.
- Comment
Lines 49-52: "There are numerous biologically active molecules with nitrogen, sulphur and oxygen heteroatoms which always drawn the attention of chemist over the years mainly because of their biological importance, therefore, extensive research is still needed to improve their properties and to reduce their adverse effects. Thiazolidin-4-ones a privileged.." could be simply changed in "Numerous heterocyclic compounds containing nitrogen, sulphur and oxygen heteroatoms have attracted the attention of chemists over the years because of their biological properties. In particular, the thiazolidin-4-ones a privileged .."
Response
We agree with the reviewer suggestion. We have changed the sentence and given in the revised manuscript.
- Comment
Lines 55-59: "...Antidiabetic [16], antiarthritic [17], FSH agonist [18], JNK stimulating phosphatase-1 (JSP56 1) [19], colon cancer (HT29) [20], breast cancer (MCF-7) [21], CNS-penetrant [22], CDK1/Cyclin B inhibition [23], COX-2 inhibitor [24] and hypoglycemia [25]. In the last few years, Kucukguzel et al. developed substituted 4-thiazolidinones among them compounds 1 and 4 exhibits excellent antitubercular activity with 90% and 98% inhibitions....." could be changed in "...antidiabetic [16], antiarthritic [17], FSH agonist [18], JNK stimulating phosphatase-1 (JSP56 1) [19], CNS-penetrant [22], CDK1/Cyclin B inhibition [23], COX-2 inhibitor [24] and hypoglycemia [25] properties. In the last few years, Kucukguzel et al. developed 4-thiazoli-dinones 1 and 4 which exhibit excellent antitubercular activity with 90% and 98% inhibitions.."
Response
We agree with the reviewer comment and suggestion. We have modified and incorporated in the revised manuscript.
- Comment
Lines 69-77: "Literature survey revealed that, sulfonamides derivatives are endowed with numerous therapeutic activities such anticonvulsants, HIV protease inhibitors, and antitumor agents [34]. It is also useful for treating certain forms of malaria, urinary tract infections, herbicides, dyes [35] and erectile dysfunction [36]. Isoxazole and its derivatives received great importance because they possess various interesting pharmacological properties. An isoxazole derivative widely exists in various natural products [37]. 5-methylisoxazole exists in many marketed drugs such as valdecoxib, parecoxib and leflunomide. Methylisoxazole
exhibits broad spectrum activity like anticancer, antiviral, antibacterial, analgesic, anticonvulsant, antinociceptive and immunomodulating activities [38]." could be changed in "Literature survey revealed that sulfonamides derivatives are endowed with numerous therapeutic activities as anticonvulsants, HIV protease inhibitors, and antitumor agents [34]. Sulfonamide derivatives are also used for treating certain forms of malaria, urinary tract infections, erectile dysfunction [36] as well as herbicides and dyes [35]. Also isoxazole derivatives possess various interesting pharmacological properties. 5-methylisoxazole is present
in many marketed drugs such as valdecoxib, parecoxib and leflunomide and exhibits broad spectrum activity like anticancer, antiviral, antibacterial, analgesic, anticonvulsant, antinociceptive and immunomodulating activities [38].
Response
We agree with the reviewer suggestion. We have changed the sentence and given in the revised manuscript.
- Comment
Lines 78-97: "The design of sulfamethaoxazole incorporated substituted 4-thiazolidinones hybrids are mainly divided into three different part as depicted in Figure 2. The 4-thiazolidinone motif is the main backbone of the design strategy. It helps to enhance the pharmacophoric properties as they exhibits drug like properties [39]. The second segment is showing sulfonamides linkages with isoxazole which is responsible for biological activity. Sulfonamides and isoxazole scaffolds are present in many reported drug such as Celecoxib and Parecoxib. Last of all, the aryl ring with the diverse substitutional unit, which is responsible for the lipophilicity control while contributing highly potent pharmacological part due to the existence of various functional groups. Considering the pharmacological importance of the above and in continuation of our efforts based on the design, synthesis and biological evaluation of heterocyclic compounds [40], herein, we have design and synthesized 4-thiazolidinone hybrids by accumulating sulfonamides linked with isoxazole unit, substituted aryl rings and mercapto group in a single molecular framework with the hope to obtain prominent antimycobactrial activity with minimal side effects." I think it is better to change in : "According to the pharmacological importance of the above reported compounds and in continuation of our efforts based on the design, synthesis and biological evaluation of heterocyclic derivatives [40], we have designed and synthesized 4-thiazolidinone hybrids by assembling, in a single molecular framework, sulfonamides and isoxazole units, with the aim to obtain prominent antimycobactrial activity with minimal side effects. The design of sulfamethaoxazole 4-thiazolidinones hybrids considers three different parts, as depicted in Figure 2. The 4-thiazolidinone motif is the main backbone of the design strategy. It helps to enhance the pharmacophoric properties of the synthesized compounds[39]. The second segment is the sulfonamide unit linked to the isoxazole nucleus which is responsible for the biological activity. Sulfonamides and isoxazole scaffolds are present in many reported drug such as Celecoxib and Parecoxib. Last of all, the aryl ring with the diverse substitutional unit, is responsible for the lipophilicity control while contributing as a highly potent pharmacological part due to the existence of various functional groups."
Response
We agree with the reviewer comment and suggestion. We have modified and incorporated in the revised manuscript.
- Comment
Lines 100-102: "We have described a protocol for the syntheses of a series of new sulfamethaoxazole incorporated substituted 4-thiazolidinone hybrids (7a-l) as a potential antitubercular agents from commercially available starting materials (Scheme 1 & 2). These compounds (7a-l)..." could be changed in "The protocol for the syntheses of a series of new sulfamethaoxazole 4-thiazolidinone hybrids (7a-l) is reported in Schemes 1 and 2. Compounds 7a-l....."
Response
We apologize for this mistake and corrected in the revised manuscript.